# A Next-Generation 3D Tissue-Engineered Model of the Human Brain Microvasculature to Study the Blood-Brain Barrier

**DOI:** 10.3390/bioengineering10070817

**Published:** 2023-07-08

**Authors:** Kalpani N. Udeni Galpayage Dona, Servio H. Ramirez, Allison M. Andrews

**Affiliations:** 1Department of Pathology & Laboratory Medicine, Lewis Katz School of Medicine at Temple University, Philadelphia, PA 19140, USA; kalpani.nisansala.udeni.galpayage.dona@temple.edu (K.N.U.G.D.);; 2Center for Substance Abuse Research, Lewis Katz School of Medicine at Temple University, Philadelphia, PA 19140, USA; 3The Shriners Hospitals Pediatric Research Center, Philadelphia, PA 19140, USA

**Keywords:** tissue engineering, blood-brain barrier, microfluidics, vasculature

## Abstract

More than a billion people are affected by neurological disorders, and few have effective therapeutic options. A key challenge that has prevented promising preclinically proven strategies is the translation gap to the clinic. Humanized tissue engineering models that recreate the brain environment may aid in bridging this translational gap. Here, we showcase the methodology that allows for the practical fabrication of a comprehensive microphysicological system (MPS) of the blood-brain barrier (BBB). Compared to other existing 2D and 3D models of the BBB, this model features relevant cytoarchitecture and multicellular arrangement, with branching and network topologies of the vascular bed. This process utilizes 3D bioprinting with digital light processing to generate a vasculature lumen network surrounded by embedded human astrocytes. The lumens are then cellularized with primary human brain microvascular endothelial cells and pericytes. To initiate mechanotransduction pathways and complete maturation, vascular structures are continuously perfused for 7 days. Constructs are validated for complete endothelialization with viability dyes prior to functional assessments that include barrier integrity (permeability) and immune-endothelial interactions. This MPS has applications for the study of novel therapeutics, toxins, and elucidating mechanisms of pathophysiology.

## 1. Introduction

The blood-brain barrier (BBB) is a highly selective interface that regulates the movement of molecular traffic bidirectionally between the central nervous system (CNS) and the cerebral blood flow [1,2,3]. Our understanding of BBB biology has improved over the decades thanks to animal studies and 2D models [4,5]. Although instructive, in vivo studies are time consuming, and species differences have reduced the translatability of results [6,7,8]. Conventional in vitro models of the BBB have been useful for exploring BBB function and CNS drug penetrability [9,10,11]. Moreover, in vitro models offer benefits in practicality and the ability to incorporate primary human cells that reflect gene and protein expression closer to that of the human brain. However, most in vitro systems utilized are planar and fail to mimic vascular properties such as perfusion and complex features of the neurovascular unit (NVU) [12]. The use of tissue engineering to design models of the BBB offers the ability to recreate tissue environments with a 3D cytoarchitectural arrangement and vascular geometries. Specifically, this can be accomplished with 3D templating, 3D bioprinting, and self-assembling-based techniques [6,13].

It has long been recognized that endothelial cells respond to the drag forces caused by fluid movement along their apical surface [14]. The importance of having a system that allows for the incorporation of fluidic flow and shear stress is underscored by the need for stimulating mechanotransduction. This mechanotransduction plays a critical role in the regulation of vessel homeostasis, as well as protein and gene expression [15]. With regard to the BBB, shear stress induces polarity and upregulates tight and adherens junction proteins, thus stimulating barriergenesis [16,17]. Furthermore, models that have combined fluidic flow and co-culture (e.g., astrocytes) demonstrate closer physiological environments that reinforce the defining properties of the BBB (low permeability, high electrical resistance) [18,19,20].

Given the physiological significance for a 3D cytoarchitecture arrangement and the presence of vessel-like fluid dynamics, we have developed a microfluidic construct that closely resembles the in vivo environment of the human brain vasculature using light-based bioprinting technology. The cells incorporated in this model are primary human brain microvascular endothelial cells (hBMVECs), pericytes, and astrocytes in a 3D hydrogel matrix. This methodology outlines the steps required to fabricate this microfluidic construct and fully vascularize the 3D lumens, which may be used in academic, government, or industry laboratories to study BBB pathophysiology. These multiphysiological systems (MPS) are suitable for research applications in drug delivery and discovery, gene therapy, toxicology, and the pathogenesis of neurological disorders.

## 2. Materials and Methods

### 2.1. Materials

The following items were used for cell culture media: Advanced DMEM/F12 (Thermo Fisher Scientific, cat. no. 12634010), EGM^TM^-2 MV Microvascular Endothelial Cell Growth Medium-2 BulletKit^TM^ (Lonza, Basel, Switzerland, cat. no. CC-3202), Gibco fetal bovine serum (Thermo Fisher Scientific, Waltham, MA, USA cat. no. 16000044), Corning™ Endothelial Cell Growth Supplement, ECGS, (Fisher Scientific, cat. no. 356006), Heparin sodium salt, derived from porcine intestinal mucosa (Thomas Scientific, cat. no. H4784-1G), penicillin-streptomycin (10,000 U/mL) (Thermo Fisher Scientific, cat. no. 15140122), Amphotericin B (Thermo Fisher Scientific, cat. no. 15290018), phosphate-buffered saline (PBS) without magnesium and calcium (Fisher Scientific, cat. no. BP3991). The bioprinting inks included Volumetric’s PEGDA PhotoInk™10 mL (Cellink, cat. no. D16110022028), PEGDA200 PhotoInk™10 mL (Cellink part of Bico, Gothenburg, Sweden, cat. no. D16110022029), PEGDA500 PhotoInk™10 mL (Cellink, cat. no. D16110022030), PEGDA Start PhotoInk™10 mL (Cellink, cat. no. D1611002260), and GelMA PhotoInk™10 mL (Cellink, cat. no. D16110022026). Extracellular matrix components and fluorescent tracers are as follows: R&D Systems™ Cultrex™ Human Fibronectin PathClear™ (Fisher Scientific, cat. no. 34-200-0101), Corning™ Collagen I, Rat (Fisher Scientific, cat. no. CB-40236), Thiol-Modified Hyaluronan/Heparin Mixture (Advanced biomatrix, Carlsbad, CA, USA cat. no. GS215), Corning^®^ Matrigel^®^ Basement Membrane Matrix (Millipore Sigma, St. Louis, MO, USA cat. no. CLS354234-1EA), Acetic Acid, Glacial (Certified ACS) ™ (Fisher Scientific, cat. no. A38S-500). CellTracker™ Green CMFDA (5-chloromethylfluorescein diacetate) (Thermo Fisher Scientific, cat. no. C7025), CellTracker™ Red CMTPX (Thermo Fisher Scientific, cat. no. C34552), and Hoechst 33,342 Solution (20 mM) (Thermo Fisher Scientific, cat. no. 62249). Reagents used for molecular biology and immunofluorescence staining: Dulbecco’s phosphate-buffered saline (DPBS) with calcium and magnesium) (Thermo Fisher Scientific, cat. no. 14040182), 16% paraformaldehyde (PFA) solution (Fisher Scientific, cat. no. AA433689L), Triton^™^ X-100 for molecular biology (Millipore Sigma, cat. no. 9036-19-5), bovine serum albumin (Millipore Sigma, cat. no. A7906-100G), Purified Mouse Anti-Human ZO-1 (Mouse, 1:200 dilution, BD Bioscience, cat. no. 610966), anti-GFP antibody (Goat, 1:200 dilution, Abcam, cat. no. ab5450), anti-CD31 antibody (rabbit, 1:100 dilution, Sino biological, cat. no. 10148-T62), anti-alpha smooth muscle actin antibody (Mouse, 1:10, Abcam, cat. no. ab7817), rhodamine phalloidin (1:1000 dilution, Cytoskeleton, Inc., Denver, CO cat. no. PHDR1), Donkey anti-Mouse IgG (H + L) Highly Cross-Adsorbed Secondary Antibody, Alexa Fluor™ 488 (1:100 dilution, Thermo Fisher Scientific, cat. no. A21202), Donkey anti-Rabbit IgG (H + L) Highly Cross-Adsorbed Secondary Antibody, Alexa Fluor™ 488 (1:100 dilution, Thermo Fisher Scientific, cat. no. A21206), Donkey anti-Mouse IgG (H + L) Highly Cross-Adsorbed Secondary Antibody, Alexa Fluor™ 594 (1:100 dilution, Thermo Fisher Scientific, cat. no. A21203), and Donkey anti-Rabbit IgG (H + L) Highly Cross-Adsorbed Secondary Antibody, Alexa Fluor™ 594 (1:100 dilution, Thermo Fisher Scientific, cat. no. A21207).

### 2.2. Cell Culture

Primary human fetal brain microvascular endothelial cells (hBMVECs) and pericytes were banked after isolation from fetal brain tissue provided by the Birth Defects Research Laboratory (Seattle, WA, USA) and validated using standardized methods as we have previously described [21]. Cultures were tested and maintained free of mycoplasma. The hBMVECs and pericytes were cultured separately in T-75 flasks coated with 50 µg/mL collagen type I. The hBMVECs were maintained in advanced DMEM/F12 growth medium supplemented with 10% heat-inactivated fetal bovine serum (FBS), endothelial cell growth supplement (ECGS; BD Biosciences), heparin (1 mg/mL; Sigma), amphotericin B (2.5 µg/mL; Invitrogen), penicillin (100 U/mL; Invitrogen), and streptomycin (100 µg/mL; Invitrogen). The hBMVECS and pericytes used for the experiments were below passage 8. The pericytes were maintained in the same media without the addition of ECGS or heparin. All cells were maintained at 37 °C with 5% CO_2_ in an incubator.

Primary human astrocytes were obtained from CelProgen. The cells were maintained in human astrocyte culture media with serum (CelProgen) at 37 °C with 5% CO_2_ until confluent. The cells were utilized under passage 10.

### 2.3. In Silico Design of 3D Vascular Network and 3D Bioprinting

STL files of the 3D cytoarchitectural arrangements with a diameter between 150 µm and 350 µm were designed and generated using Fusion 360 v.2.0.15299 (Autodesk) and Blender v.2.92.0 (Blender Foundation) software. The STL file was then imported to a Lumen X^TM^ bioprinter (CellInk). Scaffolds with 3D cytoarchitectural arrangements were printed using the cross-linkable photo bioinks polyethylene glycol diacrylate (PEGDA) and gelatin methacryloyl (GelMA). To determine whether the lumens were printed, a 1 mL syringe with a 24-gauge needle was used to perfuse 1× PBS with red dye through the printed vascular network. The printed scaffolds were rinsed by immersing in 1× PBS overnight, followed by extracellular matrix (ECM) coating the next day.

### 2.4. Endothelialization of 3D-Printed Microvessels

To introduce proper cell attachment, the printed cytoarchitectural lumen was coated with 2 mg/mL of rat tail collagen type I for 30 min at room temperature (RT). Then, the lumen was rinsed with 1× PBS and recoated with 2 mg/mL of Heprasil for 30 min at RT and rinsed with 1× PBS. The scaffolds were kept in an incubator overnight, and the next day the lumens were recoated with 2 mg/mL of matrigel and 2 mg/mL of heprasil mixture for 1 h at 37 °C with 5% CO_2_. hBMVECs were trypsinized and resuspended at 4 × 10^7^ cells/mL in 8% dextran (MW 60,000–90,000). Cells were introduced to the lumens of the scaffold using a 1 mL syringe with a 24-gauge needle, incubated at 37 °C with 5% CO_2_ and placed on an organoflow customized rocker platform (Mimetas) for 30 min. Additional cells were introduced into each scaffold and kept on a rotator (Barnstead/Thermolyne labquake) in an incubator at 37 °C with 5% CO_2_ for 30 min, and this was repeated for 4–6 h. Each scaffold was kept in a sterile 35 mm glass bottom dish during the cell seeding steps. After cell seeding, each scaffold was rinsed with culture medium to remove any cells outside the scaffold and then transferred to a new sterile 35 mm glass bottom glass bottom dish. The scaffold was covered with a PDMS cap, and the cells were cultured under perfusion at 2 µL/min for the first two days of culture. After two days, the flowrate was increased to 4 µL/min and further increased to 106 µL/min on the 6th day. For the perfusion of the microvessels, we used the equation τ = 4μQ/(πr^3^) to calculate the shear stress values, where τ is the shear stress, Q is the flow rate, r is the vessel radius, and μ = 0.007 dyne/cm^2^. The vessel diameter (maximum) was 175 μm, and a flow rate of 108 μL/min is required to achieve a physiological shear stress of 3 dyne/cm^2^. To verify the complete endothelization, the cells were stained with CellTracker™ Red CMTPX prior to cell seeding, and fluorescence images were taken from the day of seeding to day 7.

To co-culture with primary human pericytes, cells were introduced to the pre-coated lumens (as described above) at concentrations of 1 to 2 × 10^6^ cells/mL in 8% dextran (MW 60,000–90,000) using a 1 mL syringe with a 24-gauge needle and incubated at 37 °C with 5% CO_2_ and placed on an organoflow customized rocker platform (Mimetas) for 1 h. Then, primary human endothelial cells were introduced to the lumens, and cell seeding was completed as described above. Both primary human endothelial cells and pericytes were cultured under perfusion.

### 2.5. Immunofluorescence Staining

The scaffolds were fixed with 4% paraformaldehyde and permeabilized with 0.1% triton-X. Phalloidin staining was performed to visualize actin filaments in the 3D microvascular network using Acti-Stain 555 phalloidin at a 0.1 µM concentration for 30 min. To identify hBMVECs and pericytes in the lumens, the vessels were stained with anti-CD31/PECAM1 antibodies diluted at 1:100 and anti-alpha smooth muscle actin diluted at 1:100 and incubated for 2 h at RT. To visualize the tight junction protein Zonula occludenes-1 (ZO-1), the microvessels were stained with anti-ZO-1 diluted at 1:50 overnight at RT. Secondary antibodies anti-rabbit Alexa fluor 488 and anti-mouse Alexa fluor 594 were at a dilution of 1:100 for 2 h. The cell nuclei were counterstained with DAPI, and the scaffold was mounted with the ProLong antifade reagent.

### 2.6. Permeability Assay

40 kDa dextrans conjugated to tetramethylrhodamine isothiocyanate (TRITC) were used to confirm the paracellular permeability associated with a loss of barrier integrity. The endothelialized microvessels were treated with 100 ng/mL TNFα for 24 h, and then TRITC-dextran (5 mg/mL) was introduced into the lumens for 10 min. Images were captured using an EVOS M7000 fluorescence microscope at T = 0 min and 10 min at 10× magnification. Permeability was quantified using Aivia imaging software (Leica) and determined by calculating the percent permeability from the fluorescent intensity by pixel of the gel divided by that of the lumen [6]. The permeability values were compared to scaffolds not treated with TNFα.

### 2.7. Integration of Astrocytes and Immune Cells into the Scaffold

In total, 2 × 10^6^ primary human astrocytes labeled with CMPTX were suspended into the PEGDA-GelMA bioink mixture and bioprinted. Following the printing process, the lumens were endothelialzed as described in Section 2.4. Primary human monocytes from healthy donors were obtained from the University of Pennsylvania CFAR Cell and Immunology Core. The monocytes were labeled with calcein AM and then perfused through the scaffold. Images were acquired using a Nikon A1R confocal microscope and EVOS M7000 fluorescence microscope (Thermo Scientific™).

### 2.8. Statistical Analysis

The experiments were independently performed multiple times (at least 3 times for all the data shown). Student’s t-test was utilized to perform group comparisons. Statistical significance was defined as *p* < 0.05. The data collected were analyzed using prism v9.0 (GraphPad Software, San Diego, CA, USA). The imaging software Aivia v12 (Leica Microsystems, Deerfield, IL, USA) was utilized for the analysis of all images shown in the manuscript.

## 3. Results

### 3.1. Overview of 3D Scaffold Bioprinting

In this study, we developed a next-generation 3D microfluidic chip model with perfusable brain vasculature. Using light-based bioprinting technology, scaffolds were constructed and lined with primary hBMVECs and pericytes. The advantages of this model compared to alternative methods were discussed in our previously published review article [4]. The key advantages of this microfluidic construct are the ability to incorporate multiple cell types within the same hydrogel matrix, inclusion of a perfusable microvascular network, and the creation of complex cytoarchitectural. The 3D vessel network has diameters ranging from 80 µm to 350 µm. During the bioprinting process, STL files with vascular geometries were uploaded to a Lumen X^TM^ bioprinter. Bioinks were prepared by mixing PEGDA and GelMA at an 85:15 ratio. To print an ideal construct, the inks were vortexed first, then heated at 60 °C, and finally incubated at 37 °C in order to ensure a homogenous mixture. Then the bioink was loaded onto PDMS-coated dishes (Cellink) on the heated print bed. The use of PDMS dishes serves as a hydrophobic surface in which the bioinks do not adhere. Scaffolds were then generated with a 50 μm step size between polymerizing layers with UV light. The scaffold was easily removed using a plastic razor, and the gels were rinsed to remove excess dye. To visualize that the lumens of the microvascular network were open, we perfused a red-colored food dye dissolved in PBS and visualized the 3D bioprinted structure (Figure 1).

### 3.2. PDMS Scaffold Holder

To mimic the in vivo nature of the human brain vasculature, it is necessary to culture hBMVECs under physiological conditions. To achieve this, we generated a customized cap using PDMS via the soft lithography technique. This step is necessary since the gel scaffold is too soft to support coupling to the perfusion tubing without the support structure. The benefits of this type of cap are the ease of fabrication and the ability to tailor the cap to any scaffold size. First, a bioprinted scaffold with the desired dimensions was placed in an imaging dish with perfusion needles, and then uncured PDMS was poured around it. After the cap was cured in an oven, it was removed and used with a newly bioprinted scaffold (Figure 2). Furthermore, these caps are reusable after being sterilized by autoclaving.

### 3.3. Gel Structural Integrity

During the bioprinting process, we encountered several types of failures that made the scaffold unusable. This included the presence of unintended bubbles within the structure, poor stiffness, scaffold dissolution, partial polymerization, high elasticity, and cluster formation from improper mixing of the bioinks (Figure 3). To address these mechanical and technical failures, multiple approaches were taken, including vortexing (for proper bioink mixing), warming the bioink (in an oven at 60 °C for 8 min), allowing the print bed to reach maximum temperatures (set by the manufacturer), and changing the printing parameters (e.g., exposure time). During the printing process, the bioprinter parameters (exposure time 0 s–30 s, power 0 mW/cm^2^–51 mW/cm^2^, and first-layer burn-in 1×–5×) were fine-tuned depending on the bioink used. When mixing the two bioinks, these printing parameters were also adjusted. After addressing all the challenges, we found that the optimum printing parameters were a 21 s exposure time, 20 mW/cm^2^ power, and 4× first-layer burn-in to print an ideal scaffold.

### 3.4. Gel Composition and Cell Attachment

We tested multiple combinations of bioinks in order to determine the optimal composition for cellular attachment. This included PEGDA PhotoInk^TM^, PEGDA 200PhotoInk^TM^ and PEGDA 500PhotoInk^TM^. PEGDA 200PhotoInk^TM^ is stiffer and more durable than PEGDA, but is less absorbent. PEGDA 500PhotoInk^TM^ is stiffer than cartilage and loses some optical transparency properties of PEGDA. We also examined combinations with GelMA, a gelatin derivative that can improve cellular attachment. The PEGDA 500PhotoInk^TM^ optical properties prevented clear visualization of the lumens (Figure 4A,B). The stiffer gels were more likely to result in endothelial cellular clumping rather than attachment (Figure 4C). We found that the photoinks PEGDA and GelMA provided the best biocompatible environment to culture primary hBMVECs (Figure 4A–D). Additionally, combinations of rat tail collagen type I (2 mg/mL), heprasil (2 mg/mL), and matrigel (2 mg/mL) allowed the best cell attachment. Cell density also impacted proliferation and the degree of endothelialization. We found that seeding hBMVECs at 4 × 10^7^ cells/mL in 8% dextran (MW 60,000–90,000) every 30 min for 4–6 h ensured cell attachment along all channel walls of the network (Figure 4E,F).

### 3.5. Endothelialization over Time

Images of the perfused scaffold were taken daily using an EVOS FL Auto microscope (ThermoFisher Sci) housed in an environmental chamber at 37 °C, 5% CO_2_. The cells were acclimated to physiological shear stresses with a perfusion rate of 2 µL/min for the first two days, then increased to 4 µL/min for the next two to four days, and then a higher flow rate of 108 µL/min on day 5 or 6 (approximately equivalent to starting at 0.05 dyns/cm^2^ and ending at 3 dyns/cm^2^). On day 0 (Figure 5A), the entire channel network is filled with cells that have been introduced and are beginning to attach. The next day (Figure 5B), the cells are attached and begin to proliferate to fill in any gaps present (Figure 5C). By day 5 (Figure 5D), the endothelial cells are confluent and have begun contact inhibition and barriergenesis.

### 3.6. Characterization of BBB Properties

To further develop the cellular arrangement of the BBB microvascular network, pericytes were seeded (~2–4% of the amount of hBMVECs) onto the scaffold microchannels together with hBMVECs. Immunofluorescence staining was performed to confirm cell specificity and cellular arrangement. Validation of pericyte presence was confirmed using antibodies to the α-SMA (α-smooth muscle actin) protein [22] (Figure 6A). Note that pericytes naturally orient toward the abluminal side of the construct under the endothelial cells lining the lumen. For confirmation of the endothelial phenotype, antibodies to the endothelial specific marker CD31 or PECAM-1 were used, which can be seen in the cells lining the lumen (Figure 6A). Importantly, to ensure that brain endothelial cells are forming tight junctions, which are critical for limiting the paracellular movement of solutes across the BBB, antibodies to ZO-1 were used. As can be seen in the max projection (Figure 6B), ZO-1 expression was highly enriched in these cells. A 3D slice view from the lumen shows that ZO-1 expression is localized at the cell borders (Figure 6B, inset). On separate matured scaffolds, permeability assays were performed as a means to evaluate functional outcomes of barrier integrity (Figure 6C–G). Permeability evaluations utilized a fluorescently tagged dextran (of known molecular weight) that was perfused into the lumen, while images of both the lumen and parenchymal-like areas of the scaffold were taken at time 0 and at 10 min (Figure 6C–F). The amount of dextran leaking out of the lumen of a 300 μm vessel was calculated in scaffolds that were untreated or treated with TNF-α for 24 h. TNF-α, which is known to decrease barrier tightness, caused higher dextran conjugated dyes to leak out of the lumen compared to the untreated condition (Figure 6G).

### 3.7. Incorporation of Additional Cell Types into the 3D Microfluidic Construct

Aside from brain endothelial cells and pericytes, the BBB is also supported by astrocytes. Therefore, in order to advance toward a more physiological BBB structure, primary human astrocytes were introduced into the biologically compatible bioink prior to the bioprinting process (Figure 7). Note that bioinks from CellInk contain a yellow dye that protects the cells from the UV light used in the bioprinting process. Astrocytes were uniformly distributed within the gel and near the microchannels of the scaffold. Astrocytes (confirmed by expression of the astrocytic marker GFAP) can be labeled with fluorescent CellTracker dyes and visualized using fluorescence microscopy (Figure 7). Finally, since the scaffolds are fully perfusable, the system is compatible with cells found in the blood (i.e., leukocytes). Therefore, a suspension of primary human monocytes (fluorescently labeled) can be seen perfused through the scaffold. The introduction of immune cells into these 3D bioprinted scaffolds supports studies on immune-endothelial interactions (Figure 7).

## 4. Discussion

Tissue-engineered microvascular models allow investigations into human vascular function and pathology [23,24,25]. Moreover, these models have the potential to bridge the translational gap and advance the development of effective therapeutics for neurological disorders [26]. Several methods are currently being used to generate 3D models of the BBB (reviewed in [6]). A key advantage of the use of 3D bioprinting techniques is the ability to recreate advanced vascular topologies with cylindrical vessels in a biocompatible matrix [27]. Moreover, this 3D bioprinting technology is nozzle free with high resolution, which helps to control bioink bubble dynamics. It also allows a fast and continuous 3D printing process using UV or near-UV wavelength to crosslink hydrogels.

Here, we have detailed a practical technical approach that is robust and flexible for generating a 3D construct suitable for studies of the BBB. The method begins with a computer-generated vascular topology, which was then 3D printed using DLP technology (Lumen-X^TM^). Due to the versatility of this methodology, countless vascular geometry configurations can be designed to fit the application needed. Importantly, for real-time direct visualizations, we demonstrated that bioinks PEGDA + GelMA at an 85:15 ratio provided the ideal optical properties and cellular attachment for the scaffold. Moreover, proper heating and mixing of the bioinks decreased the generation of unusable scaffolding. In this advanced model of the BBB, primary human astrocytes can be embedded inside the scaffold prior to polymerization. Additionally, seeding primary human pericytes with brain microvasculature endothelial cells allows for important cellular support of the BBB. Overall, a fully endothelialized microchannel network could be generated within 5–7 days with physiological shear stresses and cytoarchitectural and cellular arrangement. Functional assays that can be performed with this MPS of the BBB include permeability measures of barrier integrity, determination of permeability coefficients of CNS drugs, efficacy testing of drug delivery platforms, immune-endothelial interaction assessments, immune infiltration analysis, and modeling vascular malformations. Compared to the other existing 3D microvascular models, this model offers several advantages, including the creation of a complex branching network, a perfusable microvascular network, and the incorporation of other brain cells surrounding the microvascular network, which will more closely resemble the in vivo nature of the human BBB.

Future directions for this work include the incorporation of functional neuronal networks into the scaffold, optimizing the interactions of cells in the biocompatible matrix, and addition of functionalized peptides [28] to improve cell attachment and spreading within the scaffold matrix. It is important to note that a current limitation for entry- to mid-level 3D bioprinting devices is the resolution of the polymerized layers, which cannot achieve capillary scale lumens. DLP-based printing on this scale can be achieved with highly expensive devices that utilize two-photon lithography [29,30].

## 5. Conclusions

This study provides a methodology for a next-generation microfluidic chip model of a 3D cerebral vascular network that can be implementable in academic, government, and industry laboratories as a preclinical platform for the study of the BBB. Moreover, these devices have great potential in the discovery of novel therapeutic approaches for neurodegenerative diseases, stroke, neuroinflammation, and congenital anomalies of the CNS.

## Figures and Tables

**Figure 1 bioengineering-10-00817-f001:**
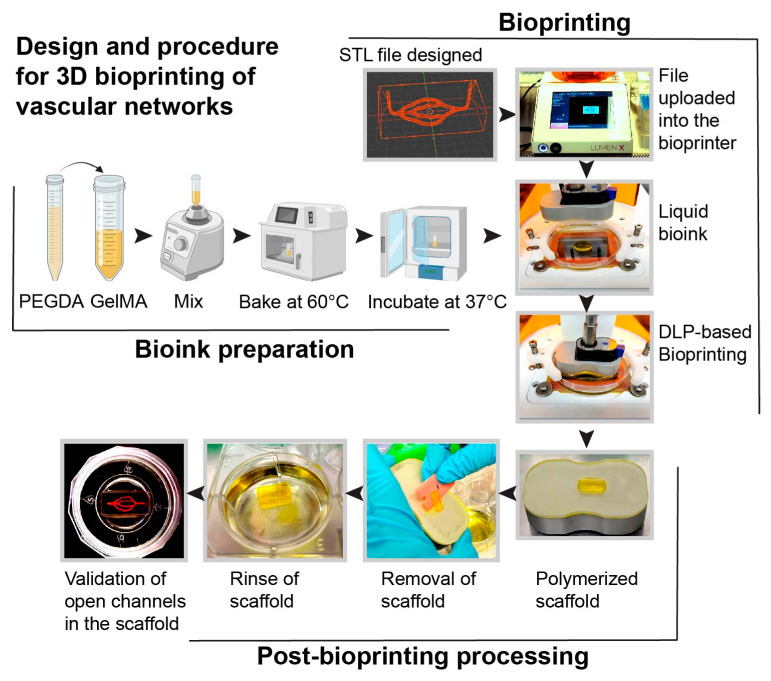
Schematic representation of the steps for the fabrication of the scaffold with a 3D vascular network. The bioprinting steps include generating a computer-aided design file (STL) of a 3D microvascular network using a computer graphic software tool (i.e., Blender) and uploading it to the bioprinter. Bioinks were prepared by mixing PEGDA and GelMA, heating at 60 °C, and then incubating at 37 °C. The liquid bioink was then polymerized using digital light-based (DLP) bioprinting to form a scaffold with channels. Red food coloring dissolved in PBS was then perfused to validate the integrity of the hollow channels.

**Figure 2 bioengineering-10-00817-f002:**
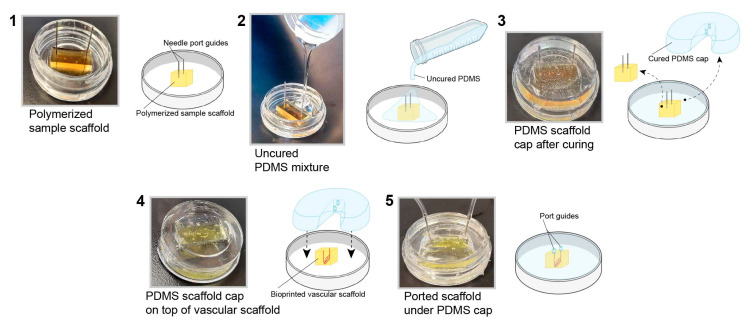
Fabrication of a PDMS cap to perfuse the 3D bioprinted scaffold. A PDMS cap was fabricated using the soft lithography technique. The completed polymerized PDMS cap was applied to the top of the scaffold to support placement of the needles for perfusion.

**Figure 3 bioengineering-10-00817-f003:**
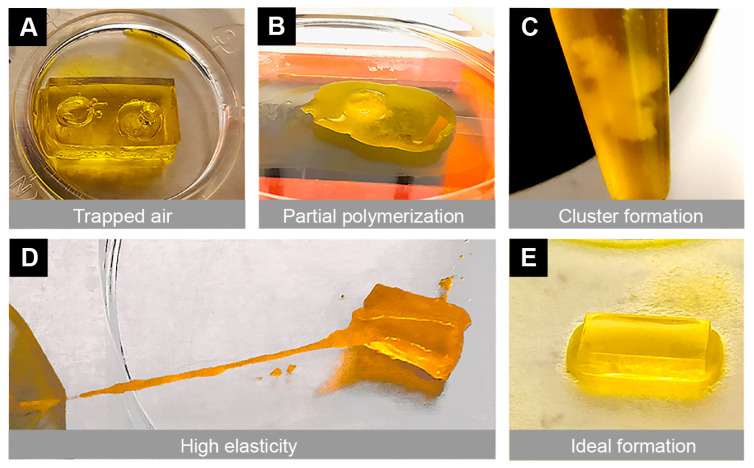
Illustration of mechanical and technical challenges that can occur during the 3D bioprinting process. Multiple challenges were encountered due to poor printing settings, which resulted in undesirable mechanical properties of the scaffold. Representative images show (**A**) air bubbles trapped in the scaffold, (**B**) incomplete polymerization of the gel, (**C**) clumping of the bioink mixtures, and (**D**) high elasticity of the scaffold. These challenges were overcome with optimized printing parameters, temperature settings, and improved bioink mixing techniques. (**E**) An ideal scaffold was printed with a 21 s exposure time, 20 mW/cm^2^ power and 4× first-layer burn-in with PEGDA + 15% GelMA after vortexing and warming in an oven at 60 °C for 8 min.

**Figure 4 bioengineering-10-00817-f004:**
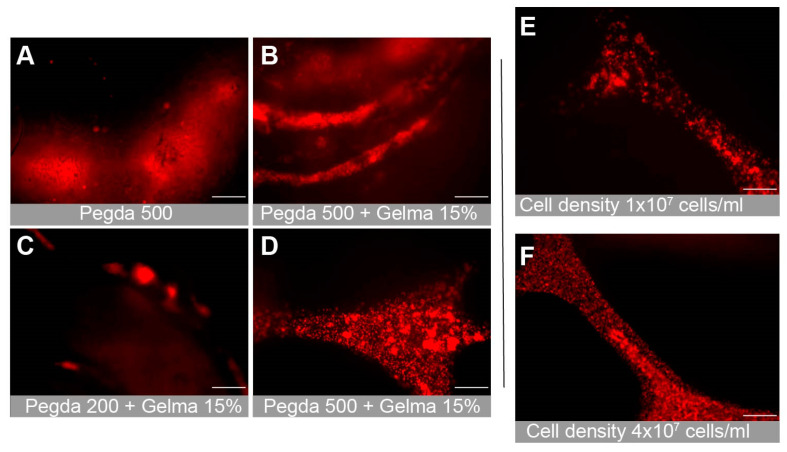
Optimization of the bioink combination for cell attachment and proliferation. Multiple combinations of bioinks were tested to determine the optimal mixture, which allowed proper cellular attachment and microscopic imaging. Representative images show (**A**) PEGDA500, (**B**) PEGDA500 + 15% GelMA, and (**C**) PEGDA200 + 15% GelMA. (**D**) PEGDA + 15% GelMA. The PEGDA + 15% GelMA mixture was found to be the optimal bioink combination. Additionally, multiple cell seeding concentrations were tested to discover the seeding density that resulted in full cell proliferation and attachment. Images show (**E**) 1 × 10^7^ cells/mL and (**F**) 4 × 10^7^ cells/mL. Scalebar equals 200 microns.

**Figure 5 bioengineering-10-00817-f005:**
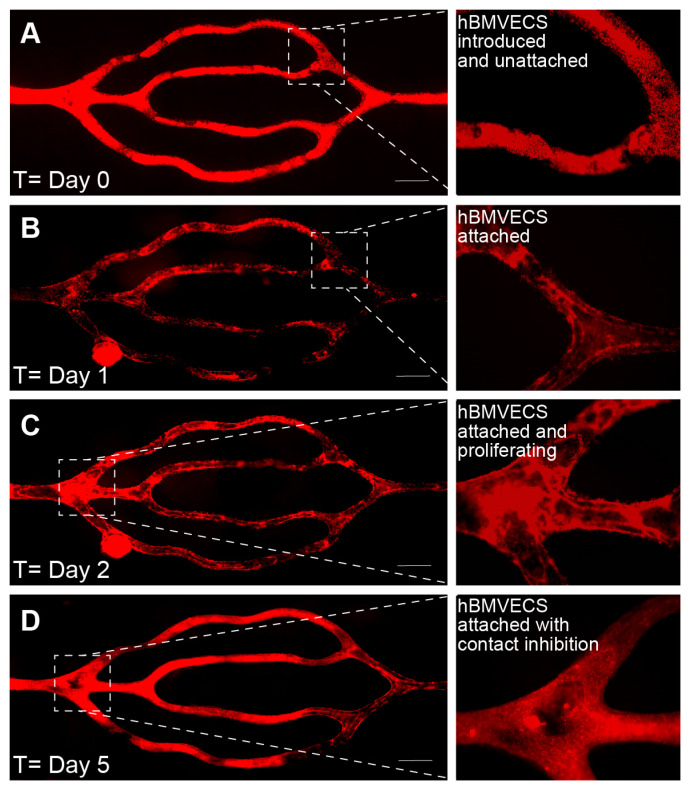
Endothelialization over time. Time series images depicting the endothelialization of lumens from a 3D bioprinted scaffold. The hBMVECs were labeled with the fluorescence dye CellTracker™ Red CMTPX. (**A**) On day 0, the cells are seen in a dextran suspension within the microchannels. (**B**) On day 1, the cells begin to attach to the walls of the microchannels but are not confluent. (**C**) On day 2, the attached cells proliferate to fill open spaces. (**D**) By day 5, the endothelial cells have lined the microchannel walls. The resulting branching microvascular network was constantly perfused such that the shear stress increased from 0.05 dyns/cm^2^ to 3 dyns/cm^2^ over 5 days. Scalebar equals 500 microns.

**Figure 6 bioengineering-10-00817-f006:**
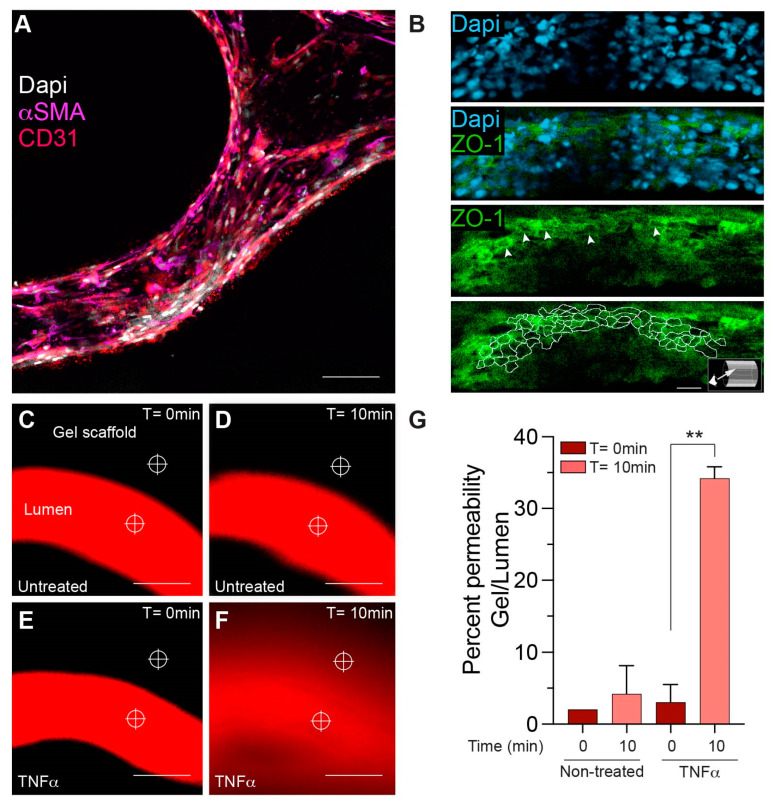
Characterization of BBB properties in a bioprinted microchannel scaffold construct. Representative images show positive immune reactivity for pericyte and brain endothelial cellular markers. (**A**) Pericytes are labeled with α-SMA (magenta), hBMVECs are labeled with the endothelial marker CD31 (red), and nuclei are labeled with DAPI (white). (**B**) Slice view of immunofluorescence labeling confirmed the junctional presence of the tight junction protein, ZO-1 (green, white arrowheads), and the nuclei were identified with dapi (blue). Aivia software image tracing was used to outline ZO-1 expression at the cell borders. Scalebar equals 25 microns. (**C**–**F**) The BBB integrity was monitored using fluorescently labeled dextrans of defined molecular weight perfused in the lumens of scaffolds treated with or without TNF-α for 24 h. (**G**) Permeability was determined using image intensity analysis per area at the reticles shown. TNF-α showed a statistically significant increase in labeled dextran permeability out of the lumen (n = 3 Student’s *t*-test ** *p* < 0.01). Scalebar equals 200 microns.

**Figure 7 bioengineering-10-00817-f007:**
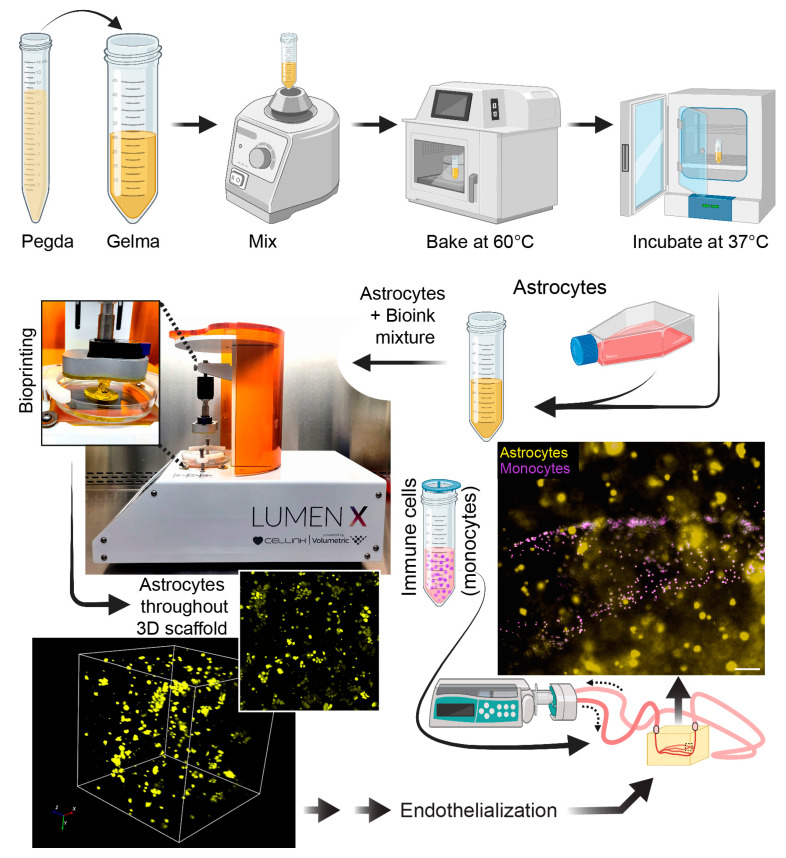
Introduction of additional cell types to the scaffold. For incorporation of astrocytes, PEGDA and GelMA were vortexed, then heated at 60 °C, and then incubated at 30 °C. After this process, which ensures a homogenous bioink mixture, 2 × 10^6^ primary human astrocytes were introduced into the gel and acclimated to 37 °C. The scaffold was then printed with the astrocytes uniformly distributed in the construct. The lumens were then endothelialized as indicated previously. Lastly, fluorescently labeled monocytes were perfused for studies relevant to immune-endothelial interactions. Scalebar equals 100 microns.

## Data Availability

All data are contained within the article.

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
