# Peer review of "A Next-Generation 3D Tissue-Engineered Model of the Human Brain Microvasculature to Study the Blood-Brain Barrier"

_bioengineering, 2023, doi:10.3390/bioengineering10070817_

Round 1

Reviewer 1 Report

This study introduces a novel 3D tissue engineered microfluidic model which creates a physiologically relevant blood-brain barrier (BBB) imitation that incorporates astrocytes within its 3D construct and perfuses both endothelial cells and pericytes to form the microvascular channels. With the fabrication of the model, several biological interactions can be simulated for a better representation of the vital properties presented by the BBB. By establishing the best bioink combination for 3D bioprinting required to generate reproducible experiments, this model offers a potential platform for investigating drug delivery dynamics and several disease states that are implicated through a malfunctioning BBB. There are several limitations present in the current manufacturing methods: the size of the vessels are larger than capillaries for the BBB, the permeability seems to be much higher than in vivo, and the cells in the parenchyma have not been thoroughly investigated.

The study offers a well-established platform, but is lacking in consistency for the cell densities and coating solutions introduced throughout the study. Many results can easily be placed under different sections, while other results can be expanded upon to showcase the novelty of the research conducted. Figures/ figure captions were subpar and need to be checked over for consistency with the rest of the paper and other figures. Overall, an exciting new platform but requires more detail and cohesive editing throughout the paper.

Minor Comments:

● Review document for grammatical errors such as missing commas, consistency in capitalization, and awkward sentence structure

● Section 2.1 & 2.3 - Materials and Methods did not discuss details on the pericytes cell line used along such as where they were received from, cultured to certain passage with what media, and how they were introduced into the final 3D printed product, etc. The first mention of pericytes in the materials and methods is in the immunofluorescence section and details on how many pericytes are added at 2-4% of the hBMVECs in the results section 3.6 are too late into the manuscript to allow the reader to understand the set-up.

● Lines 142-144: Should discuss the reasoning for this jump in flow rate to go from 0.05 to 3.0 dyns/cm^2. As well, could you provide more dimensions of the scaffold to identify the connection for how 106 uL/min equates to 3 dyns/cm^2?

● Line 172 - How many monocytes were perfused through the chip and stained with calcein AM?

● Lines 246-248: The coatings and their concentrations(specifically for Heprasil) do not match what is discussed in section 2.3.

● Figure 4 - PEGDA and GelMa does not match its spelling within the text

● Figure 4 caption - It does not discuss figure D. In addition, Figure B and Figure D relay the same information where one figure can be removed to emphasize a better representative image of the system.

● Figure 6 caption - The letters signifying which image is being described should be at the beginning of the sentence and not at the end of the sentence. As well, they do not need to be repeated for every sentence that the description is detailing i.e. (C-F)

● Figure 7 - Displaying the astrocytes and monocytes is fairly small while having cells stained in similar colors which makes it difficult to decipher between the cell types.

● Line 32: Citation [4,5] goes before the period.

● Line 123: When determining if the lumens were printed, was the PBS combined with a dye to see the network or was outflow used as the only metric? In the results, you state the use of red dye; this should also be in the methods.

● Line 134: Cells were introduced syringed into the scaffold every 30 minutes for 4-6 hours? Does this mean new cells were added each time (so 8 - 12x total).

Major Comments:

● The results section 3.1 sounds repetitive of the materials and methods section of 2.2 with a few added details such as step size of polymerization, scaffold removal, and microvascular network visualization. What result is being conveyed here so that this section cannot be absorbed into section 2.2 along with its figure (Figure 1)?

● Section 3.2 is similar to the problem listed above where it seems that this can be moved to the materials/methods section.

● Section 3.3 displays multiple challenges that needed to be addressed for the bioprinting process to overcome to be successful, but does not display what exact printing parameters were successful. At its current state, this section can be interwoven into the discussion along with its figure (Figure 3).

● Line 160: Why was 40kDa chosen as the particle size? The in vivo BBB is impermeable to sizes > 500 Da. While it is very difficult to replicate in vivo barrier tightness, using a smaller molecule may be more representative? Or justification should be given.

● Line 182. I think the advances and advantages of this platform should be discussed briefly, instead of only citing the previous work. We need appropriate, standalone context for the current results.

● Figure 1. It is difficult to see the smaller images within Figure 1. What is the vascular structure supposed to look like. There should be a larger 3D render of the intended vasculature.

● Figure 6B: Was Zo-1 localized at the cell boundaries. I've often see it before where the entire cell is somewhat green (especially with polyclonal antibodies), with increased expression at the cell-cell junctions and nucleus. From the image, there does not seem to be clear tight junction formation, and I do not think the green expression alone supports this. Maybe another stain (VE-Cadherin) would be more clear, or a western blot for the protein.

● What was the purpose of introducing monocytes or astrocytes. No communication or affect on the BBB model was shown. From the small images in Figure 7, it appears the monocytes are localized in the vessels, with no migration through the BBB? If monocytes are only being perfused alone with the media in and out of the device, then I do not understand the result.

● Other microplatforms have been investigating self assembling microvascular networks, where endothelial cells are allowed to migrate into the perynchema-mimicking scaffold and form vessels. They often require PDMS/plastic channels to create an initial seperation of endothelial cells and the scaffold. This would be interesting as a step further, where the microfluidic devices/channels themselves are made of scaffolds mimicking blood vessels. Does this metho allow for self assembling and angiogenesis into the bioprinted mimic.

Some of the parts need to be re-written to be clear the first time they are presented. 

Author Response

We are grateful to the reviewer’s comprehensive review and suggestions for this study.  The feedback has certainly improved the clarity and quality of the manuscript presentation.

Minor Comments:

  1. Review document for grammatical errors such as missing commas, consistency in capitalization, and awkward sentence structure.

We have revised the manuscript to correct for consistency, and other grammatical errors.

  1. Section 2.1 & 2.3 - Materials and Methods did not discuss details on the pericytes cell line used along such as where they were received from, cultured to certain passage with what media, and how they were introduced into the final 3D printed product, etc. The first mention of pericytes in the materials and methods is in the immunofluorescence section and details on how many pericytes are added at 2-4% of the hBMVECs in the results section 3.6 are too late into the manuscript to allow the reader to understand the set-up.

We have now included the requested description (regarding source, isolation and culturing details) in the methods section for “Cell Culture”. Additionally, section 2.3 now includes details for the pericytes in the 3D microvessels.

  1. Lines 142-144: Should discuss the reasoning for this jump in flow rate to go from 0.05 to 3.0 dyns/cm^2. As well, could you provide more dimensions of the scaffold to identify the connection for how 106 uL/min equates to 3 dyns/cm^2?

The purpose of increasing the flow rate is to apply shear fluidic forces to the endothelial cells akin to physiological conditions.  Shear stresses range from 10-70 dyn/cm2 in arterial vessels and 1-6 dyn/cm2 in venous vasculature (Papaioannou International J. of Cardiology 113, p12-18, 2006).

We revised the manuscript as follows:

The equation t=4mQ/pr^3 to calculate the shear stress values.  Where  is the shear stress, Q is the flow rate, r is the vessel radius and m=0.007 dyn·s/cm^2, and vessel diameter (maximum) is 175 microns.  In order to arrive at 3 dyn/cm^2 our flow rate must be 108mL/min.

  1. Line 172 - How many monocytes were perfused through the chip and stained with calcein AM?

We stained 5 million monocytes with calcein AM and then perfused them through the construct.

  1. Lines 246-248: The coatings and their concentrations (specifically for Heprasil) do not match what is discussed in section 2.3.

We apologize for the typo and have revised the manuscript accordingly.

  1. Figure 4 - PEGDA and GelMa does not match its spelling within the text.

Thank you. We have revised the figure and text.

  1. Figure 4 caption - It does not discuss figure D. In addition, Figure B and Figure D relay the same information where one figure can be removed to emphasize a better representative image of the system.

We apologize for the error. Figure 4D should have been labeled PEGDA + 15% GelMA and not PEGDA500 + 15% GelMA.  We have revised the manuscript accordingly.

  1. Figure 6 caption - The letters signifying which image is being described should be at the beginning of the sentence and not at the end of the sentence. As well, they do not need to be repeated for every sentence that the description is detailing i.e. (C-F).

We have revised the figure legend accordingly.

  1. Figure 7 - Displaying the astrocytes and monocytes is fairly small while having cells stained in similar colors which makes it difficult to decipher between the cell types.

Thank you for the suggestion.  The image of the astrocytes and monocytes is much larger now and also revised to show the two cell types in different colors (monocytes in magenta and astrocytes in yellow).

  1. Line 32: Citation [4,5] goes before the period.

We have revised the text.

  1. Line 123: When determining if the lumens were printed, was the PBS combined with a dye to see the network or was outflow used as the only metric? In the results, you state the use of red dye; this should also be in the methods.

We have revised the text to include the requested details under Method section 2.2. We used the PBS with red dye to confirm that the lumens were open and to see the status of the network during the optimization process (i.e printing settings and bioink combinations).  Once the above was performed, we only used PBS to perfuse through the lumens.  

  1. Line 134: Cells were introduced syringed into the scaffold every 30 minutes for 4-6 hours? Does this mean new cells were added each time (so 8 - 12x total).

 Yes.  Every 30 minutes new cells were added.

Major Comments:

  1. The results section 3.1 sounds repetitive of the materials and methods section of 2.2 with a few added details such as step size of polymerization, scaffold removal, and microvascular network visualization. What result is being conveyed here so that this section cannot be absorbed into section 2.2 along with its figure (Figure 1)?

The figure 1 is a representative overview of the 3D bioprinting process showing the key steps of recreating the 3D brain vasculature via light-based 3D bioprinting technology.  Whereas the information on section 2.2, shows the exact steps (in order) that were used to successfully bioprint a scaffold with the 3D microvasculature.

  1. Section 3.2 is similar to the problem listed above where it seems that this can be moved to the materials/methods section.

We agree with the reviewer but feel that maintaining the current format helps in the clarity and reproducibility of these methods by other laboratories. Additionally, the format is similar to other manuscripts published in the journal.  The section emphasizes the importance that in order to mature barrier endothelium, it is necessary to culture the hBMVECs under physiological conditions at all times. Thus, to achieve this, we generated a customized cap using PDMS via the soft lithography technique to allow continuous perfusion of the lumens within the scaffold.  

  1. Section 3.3 displays multiple challenges that needed to be addressed for the bioprinting process to overcome to be successful, but does not display what exact printing parameters were successful. At its current state, this section can be interwoven into the discussion along with its figure (Figure 3).

We have now included an additional image in the figure to show what we consider to be an ideal scaffold and the settings for the printer in the legend.

  1. Line 160: Why was 40kDa chosen as the particle size? The in vivo BBB is impermeable to sizes > 500 Da. While it is very difficult to replicate in vivo barrier tightness, using a smaller molecule may be more representative? Or justification should be given.”

We have tested multiple fluorescent tracers to evaluate the permeability of this construct.  Using primary brain endothelial cells, we have also previously published on permeability evaluations with various size tracers (Andrews et al. J Cereb Blood Flow Metab doi: 10.1177/0271678X17708690.).  Of note, many in-vivo evaluation of BBB permeability use 40 kDa size tracers and more commonly the larger size of 70 kDa, as it is the case when Evan’s blue binds albumin.  Please see the following references below:     

-In Di Marco, A et al study, a 4 kDa dextran shows a similar permeability kinetics similar to the 40 kDa (with the lower molecular weight showing a greater degree of variability).

Di Marco, Annalise, et al. "Application of an in vitro blood–brain barrier model in the selection of experimental drug candidates for the treatment of Huntington’s disease." Molecular Pharmaceutics 16.5 (2019): 2069-2082.

The following studies have used molecular weight tracers that are larger than 40 kDa to study BBB permeability.

-Mazzucco, M. R., Vartanian, T., & Linden, J. R. (2020). In vivo Blood-brain Barrier Permeability Assays Using Clostridium perfringens Epsilon Toxin. Bio-protocol, 10(15), e3709-e3709.

Park, Jun Sung, et al. "Establishing co-culture blood–brain barrier models for different neurodegeneration conditions to understand its effect on BBB integrity." International Journal of Molecular Sciences 24.6 (2023): 5283.

-Na Pombejra, S., Salemi, M., Phinney, B. S., & Gelli, A. (2017). The metalloprotease, Mpr1, engages AnnexinA2 to promote the transcytosis of fungal cells across the blood-brain barrier. Frontiers in cellular and infection microbiology, 7, 296.

  1. Line 182. I think the advances and advantages of this platform should be discussed briefly, instead of only citing the previous work. We need appropriate, standalone context for the current results.”

As suggested, we now include key advantages of the model in the discussion.

  1. Figure 1. It is difficult to see the smaller images within Figure 1. What is the vascular structure supposed to look like. There should be a larger 3D render of the intended vasculature.”

We have revised the figure to show a larger 3D rendering of the lumen structural geometry.

  1. Figure 6B: Was Zo-1 localized at the cell boundaries. I've often see it before where the entire cell is somewhat green (especially with polyclonal antibodies), with increased expression at the cell-cell junctions and nucleus. From the image, there does not seem to be clear tight junction formation, and I do not think the green expression alone supports this. Maybe another stain (VE-Cadherin) would be more clear, or a western blot for the protein.

Figure 6B shows the max projection of the ZO-1 which can obscure the visualization of individual cellular staining patterns. We have now included a 3D slice from the perspective of the lumen (Figure 6B inset) which shows that the ZO-1 staining is junctional.

  1. What was the purpose of introducing monocytes or astrocytes. No communication or affect on the BBB model was shown. From the small images in Figure 7, it appears the monocytes are localized in the vessels, with no migration through the BBB? If monocytes are only being perfused alone with the media in and out of the device, then I do not understand the result.

We apologize for the lack of clarity.  The purpose of figure 7 is to show the versatility and compatibility of the model with 1) embedding of cells in the gel/scaffold matrix such as astrocytes (as shown) but also microglia and neurons (not shown in this study) and 2) the addition of immune cells in the lumen compartment.  The representative image is that of monocytes traveling in the luminal flow.  The goal is to invite the reader to envision studies related to neuroinflammation where this model would facilitate evaluations on immune-endothelial interactions (i.e., crawling, adhesion, transendothelial migration etc.).

  1. Other microplatforms have been investigating self-assembling microvascular networks, where endothelial cells are allowed to migrate into the perynchema-mimicking scaffold and form vessels. They often require PDMS/plastic channels to create an initial separation of endothelial cells and the scaffold. This would be interesting as a step further, where the microfluidic devices/channels themselves are made of scaffolds mimicking blood vessels. Does this method allow for self-assembling and angiogenesis into the bioprinted mimic.

Indeed, there is a lot of potential to combine areas in the scaffold which support self-assembling microvascular networks that can connect with the larger bioprinted vessel lumens. We have ongoing projects to advance the support of various types of vascular networks.  

Reviewer 2 Report

The authors show an optimized 3d bio-ink printing method that can simulate the fluid mechanics of the blood-brain barrier.

The astrocytes are uniformly distributed into a liquid or gel and 3d printed to form a matrix. The 3d printer uses UV light to make the printout. Is there any study on the viability of cells exposed to certain levels of UV?

The authors successfully created a vasculature and completed it after seven days. Is there any study of viability or continuation of the system for extended times or months?

Figure 2 can be made more horizontal with two columns

Figure 4 is disorganized. There is no mention of part D and whether it should be the same as B. Was it a different condition?

There could be a mention of the costs of operating this bioprinting platform. Is in 10,000 USD or 50,000 USD range?

Nice work. Microfluidics with gels is tricky.

Author Response

We greatly appreciate the reviewer’s comments and praise for our study. Please see below regarding answers to the reviewer’s questions.

  1. The 3D printer uses UV light to make the printout. Is there any study on the viability of cells exposed to certain levels of UV?

Yes, multiple studies have shown the effect on cell viability after light-based (UV) 3D bioprinting technology.  Please see following references.  These evaluations have shown that an expected 90% cell viability is expected after exposure to UV (405 nm) light.  It is important to note that the yellow coloring of the scaffold helps protect the cells embedded and minimizes the light-scatter of the UV.

-Xu, He-Qi, et al. "A review on cell damage, viability, and functionality during 3D bioprinting." Military Medical Research 9.1 (2022): 70.

-Xu, Heqi, et al. "Effects of Irgacure 2959 and lithium phenyl-2, 4, 6-trimethylbenzoylphosphinate on cell viability, physical properties, and microstructure in 3D bioprinting of vascular-like constructs." Biomedical Materials 15.5 (2020): 055021.

-Adhikari, J., Roy, A., Das, A., Ghosh, M., Thomas, S., Sinha, A., ... & Saha, P. (2021). Effects of processing parameters of 3D bioprinting on the cellular activity of bioinks. Macromolecular Bioscience, 21(1), 2000179.

  1. Figure 2 can be made more horizontal with two columns.

We have revised the figure as suggested.

  1. Figure 4 is disorganized. There is no mention of part D and whether it should be the same as B. Was it a different condition?

We apologize for this oversight.  Figure 4D should have been labeled PEGDA + 15% GelMA and not PEGDA500 + 15% GelMA (the figure has been corrected).

  1. There could be a mention of the costs of operating this bioprinting platform. Is in 10,000 USD or 50,000 USD range?

Thank you for the suggestion, however we would like to leave cost information out of the manuscript since it may be misleading due to regional market differences, ongoing offers, changes due to inflation, price point differences that reflect industry sector (i.e academia vs private vs government etc) and introduction of new product lines by companies.  Thus, due to the above factors pricing can range widely.  At the time these studies were performed a Lumen X bioprinter costs approximately $45,000 USD.  An approximate cost of $50 is estimated to print each scaffold and $25 for additional supplies (needles, tubing, PDMS, etc).
